# Anti-poverty policy and health: Attributes and diffusion of state earned income tax credits across U.S. states from 1980 to 2020

Kelli A. Komro[1]*, Phenesse Dunlap[1], Nolan Sroczynski[1], Melvin D. Livingston[1], Megan A. Kelly[2], Dawn Pepin[2], Sara Markowitz[3], Shelby Rentmeester[1], Alexander C. Wagenaar[1]

**1** Department of Behavioral, Social, and Health Education Sciences, Rollins School of Public Health, Emory University, Atlanta, Georgia, United States of America, **2** Policy Research, Analysis, and Development Office, Office of the Associate Director for Policy and Strategy, Centers for Disease Control and Prevention, Atlanta, Georgia, United States of America, **3** Department of Economics, Emory University, Atlanta, Georgia, United States of America

* kkomro@emory.edu

**Data Availability Statement:** All relevant data are within the paper and its Supporting Information files.

## Abstract

### Purpose

The U.S. federal Earned Income Tax Credit (EITC) is often considered the most effective antipoverty program for families in the U.S., leading to a variety of improved outcomes such as educational attainment, work incentives, economic activity, income, and health benefits for mothers, infants and children. State EITC supplements to the federal credit can significantly enhance the magnitude of this intervention. In this paper we advance EITC and health research by: 1) describing the diffusion of state EITC policies over 40 years, 2) presenting patterns in important EITC policy dimensions across space and time, and 3) disseminating a robust data set to advance future research by policy analysts and scientists.

### Methods

We used current public health law research methods to systematically collect, conduct textual legal analysis, and numerically code all EITC legislative changes from 1980 through 2020 in the 50 states and Washington, D.C.

### Results

First, the pattern of diffusion across states and time shows initial introductions during the 1990s in the Midwest, then spreading to the Northeast, with more recent expansions in the West and South. Second, differences by state and time of important policy dimensions are evident, including size of credit and refundability. Third, state EITC benefits vary considerably by household structure.

**Funding:** The National Institute on Minority Health and Health Disparities, National Institutes of Health (https://www.nimhd.nih.gov) through award R01MD010241 to KAK and ACW, The National Heart, Lung, and Blood Institute, National Institutes of Health (https://www.nhlbi.nih.gov) through training award T32 HL130025 to PD, and the Policy Research, Analysis, and Development Office; Office of the Associate Director for Policy and Strategy; Centers for Disease Control and Prevention (https://www.cdc.gov/policy/about/index.html) supported this research. The findings and conclusions of this paper are solely the responsibility of the authors and do not necessarily represent the official position of the National Institutes of Health or the Centers for Disease Control and Prevention. The funders had no role in study design, data collection and analysis, decision to publish, or preparation of the manuscript.

**Competing interests:** The authors have declared that no competing interests exist.

## Conclusion

Continued research on health outcomes is warranted to capture the full range of potential beneficial effects of EITCs on family and child wellbeing. Lawyers and policy analysts can collaborate with epidemiologists and economists on other high-quality empirical studies to assess the many dimensions of policy and law that potentially affect the social determinants of health.

## Introduction

Across the world, health organizations acknowledge family economic conditions as an important social determinant of health [1, 2]. In the United States, the relationship between income and health operates on a continuum, with the most economically disadvantaged experiencing the poorest health outcomes [3, 4]. Living in poverty has negative implications across the life-span and across many health outcomes [3, 5, 6]. Despite substantial evidence of poverty's deleterious health effects, the economic gap between the wealthiest and poorest families continues to widen and health disparities persist [4]. Public policies aimed at reducing poverty have been shown to raise low incomes [7–9]. State governments are uniquely positioned to address state-level income inequality that has been linked with poor health outcomes [8, 10–15]. The state-level earned income tax credit (EITC) is a promising policy strategy that has been used to reduce poverty and improve related outcomes, including health [16–19].

The U.S. federal EITC was first introduced in 1975 as a tax credit to help relieve tax burdens and supplement incomes of low-earning, working families [20]. Single mothers with dependent children were the initial intended recipient population, and the EITC was expanded during welfare reform in the early 1990s to facilitate the transition from the existing welfare system into the workforce [20]. Since its introduction, modifications to the federal EITC policy include provisions for accommodating the number of children in a household, extending eligibility to childless workers, considering presence of a spouse, and adjusting for inflation [20]. The value of the federal EITC is determined by a combination of amount of earned income and family structure, and is fully refundable. Refundability means that if tax liability falls to zero and there is still some portion of the credit remaining, the government will send a refund check equal to that remaining amount to the worker [17, 19]. By design, the EITC primarily benefits working families; workers with children receive a much larger credit than workers without qualifying children. In the 2020 tax year, the maximum federal credit for families with one child is $3,584, while the maximum federal credit for families with three or more children is $6,660 [21]. In contrast, a childless worker between the ages of 25 and 65 (filing singly or, if married, jointly) can receive a maximum federal earned income tax credit of $538 [21]. Many economists consider the federal EITC as the largest and most effective antipoverty program for families in the United States [22]. Additionally, state EITC supplements to the federal credit can significantly enhance the magnitude of this intervention [23], and a few states began to introduce EITCs in the late 1980s. As of 2020, 29 states and the District of Columbia offer their own EITCs, which vary widely in eligibility, credit amount, and refundability [24].

While research on economic outcomes of the federal EITC dates back to the initial implementation of the program, much of the evidence on health outcomes has emerged within the last decade. Research has associated the federal EITC expansions with improved overall self-reported health and positive mental health outcomes among adults [25, 26], but finds conflicting results regarding effects on adult health behaviors [27–30]. The research is more consistent

regarding beneficial effects of federal EITC expansions on infant and child health. An increase in the federal EITC is associated with decreased infant mortality, higher birth weights and improved overall perinatal health [27, 30]. In addition, one study found that the EITC was associated with increased supportive home environments for child development [28] and two studies estimated that parental receipt of federal tax credits was associated with a child's increased likelihood of college enrollment and higher lifetime earnings [31, 32]. The EITC may also be relevant to reducing risk factors for child abuse and maltreatment (e.g., poverty, maternal stress and depression) [33].

The body of literature on effects of state EITCs on health outcomes has grown as more states have adopted their own policies. These studies find similar results to those assessing effects of changes to the federal EITC. A study by Muennig and colleagues concludes that a state EITC is a cost-effective policy for increasing health-related quality of life and reducing mortality [33]. Similar to studies that examine federal EITC expansions, those measuring effects of state-level credits report inconsistent findings regarding effects on adult health, including smoking and obesity [34–36]. Findings from studies examining state EITC effects on infant and child health have been more consistent [37]. State EITC introductions are associated with fewer low-birth-weight births and increased gestation weeks [23, 34, 38–40], as well as improvement in overall child health [41]. Studies also find state EITCs are associated with improved educational outcomes at primary, secondary, and post-secondary education levels [42, 43].

Given numerous expansions and changes over time in specific EITC policy features at the state level, continual policy surveillance to track those changes can facilitate further research on the health effects across the lifespan [44]. The extant literature suggests that widespread state EITC adoption appears likely to have important health benefits, especially among infants and children [19, 30–32, 35–38]. Similarly, given the current evidence regarding the beneficial effects of EITC on educational outcomes [39, 40], researchers can further contribute to the available EITC literature by assessing potential effects of varied EITC policies on reduction of intergenerational poverty. Studies of state-level policy changes are able to include important design elements, such as multiple comparison groups across space and time, that can be used to generate unbiased estimates [45]. Careful legal research and coding combined with science-based measurement methods are required to establish reliable and valid indicators of policy prevalence, timing, and dose. Research has shown that reliance on policy indicators from websites maintained by administrative agencies or advocacy organizations often include measurement error [46]. In order to scientifically evaluate health effects of state-level EITC policies, our multidisciplinary team of epidemiologists, economists, lawyers, and statisticians conducted rigorous surveillance of these policies over time and across states. In this paper we seek to advance EITC and health research by: 1) describing the diffusion of EITC policies across the U.S. states over 40 years, 2) presenting patterns in important EITC policy dimensions (such as credit amount and refundability) across space and time, and 3) disseminating a robust data set that can be used to advance future research by other policy analysts and scientists and to inform decision making by stakeholders seeking to improve a variety of outcomes.

## Methods

We used the public health law research method known as legal epidemiology [44, 47] to systematically collect, conduct textual legal analysis, and numerically code all EITC legislative changes from 1980 through 2020 in the 50 states and Washington, D.C. (collectively, the states). The Emory authors funded and worked with a team of legal researchers at Temple University's Public Health Law Research Program. This legal team began the legal analysis with a sample of five states, producing policy memos that detailed trends, variations, and key features

of laws from the initial sample. Key search terms drawn from the literature review included "tax credits" with "earned income," "low income," "working families," "family," and "refundable." Additional terms were included based on the table of contents of the tax section in each state's codified law. Based on this preliminary legal research and the published literature, the authors worked closely with the legal team to develop a policy surveillance codebook and a detailed coding protocol to capture important EITC policy dimensions, including eligibility criteria, as well as the amount and refundability of the tax credit (see S1 and S2 Appendices, respectively, available online).

The legal team then reviewed the text of all U.S. federal and state laws (statutes and regulations) on EITCs from all 50 states and Washington, D.C., using the Westlaw and LexisNexis legal databases and session laws from each state's legislative website. The search terms included: earned income tax credit, earned income credit, low-income tax credit, working families tax credit, family tax credit, refundable tax credit, section 32 Internal Revenue Code. The study includes all laws outlining eligibility and benefit information on state earned income tax credits aimed at providing tax relief or refunds to low- or moderate-income working taxpayers. For the purposes of this study, laws providing credits or other income supports to low-to moderate-income workers for assistance with food, housing, or childcare were excluded.

The initial legal coding was conducted in 2013, with updates in 2015 and 2016. The legal data was collected and coded with extensive quality control procedures, including blinded independent coding of a 20% random sample of items by two trained legal researchers. As part of Temple's work, all legal coders were closely supervised by a senior attorney, who reviewed protocols with coders for any variable showing 5% or higher cross-coder disagreement rate. Divergence rates were below 10% for the original coding and 2015 and 2016 updates. All divergences between two coders were resolved by the supervising attorney after meeting with the two coders and examining the original legal text. Following completion of the legal coding, quality control checks included inspecting descriptive data and comparing results with other EITC sources. If discrepancies were noted, the legal team returned to the original legal text to verify or update the data set. The legal team compared their results with a similar study conducted by NCSL available at: http://www.ncsl.org/research/labor-and-employment/state-earned-income-tax-creadit-enactments.aspx; https://users.nber.org/~taxsim/state-eitc.html. Any discrepancies were resolved. In 2020, a team of legal researchers at CDC's Policy Research, Analysis, and Development Office conducted an update to a subset of questions relevant to this research. They followed the methods in the protocol and outlined above with minor variations (see S5 Appendix, available online). The CDC legal researchers updated the coding for a subset of questions as developed. The researchers expanded the search terms to reflect the current state of the laws based upon an independent review of literature and sample searches. All laws were collected using the database WestlawNext. Every law was independently coded by two legal researchers and de minimis discrepancies were resolved through consensus. During this process, some anomalies were identified in data related to these variables that had been included in the original dataset. The 2020 update expanded to include a retrospective quality control review and correction, as needed, of previously coded data related to these variables. This research made use of publicly available data sets and was determined to be exempt by the Emory University Institutional Review Board.

## Results

### Dimensions of state EITC laws

Table 1 includes the core legal data set, stratified by state and time from 1980 through 2020 tax year, including effective tax year(s) of introduction and amendments, credit amount (i.e.,

**Table 1. State EITC as a percent of the federal EITC and refundability, 1980–2020.**

| State | Tax Year | Percent of federal EITC | If different by number of dependents | | | | Refundable | Legal citation |
|---|---|---|---|---|---|---|---|---|
| | | | None[a] | One[a] | Two[a] | Three[a] | | |
| AL | 1980–2020 | None | | | | | | |
| AK[b] | 1980–2020 | None | | | | | | |
| AZ | 1980–2020 | None | | | | | | |
| AR | 1980–2020 | None | | | | | | |
| CA[c] | 1980–2014 | None | | | | | | |
| | 2015 | 44 | | | | | Yes | Cal Rev & Tax Code § 17052 |
| | 2016 | 43 | | | | | Yes | |
| | 2017 | 44 | | | | | Yes | |
| | 2018–2020 | 45 | | | | | Yes | 2018 Cal. Legis. Serv. Ch 52 (SB 855) |
| CO[d] | 1980–1998 | None | | | | | | |
| | 1999 | 8.5 | | | | | Yes | C.R.S. § 39-22-123 |
| | 2000–2001 | 10 | | | | | Yes | C.R.S. § 39-22-123.5 |
| | 2002–2004 | 10* | | | | | | |
| | 2005–2010 | 0 | | | | | | |
| | 2011–2014 | 10* | | | | | | |
| | 2015–2020 | 10 | | | | | Yes | |
| CT | 1980–2010 | None | | | | | | |
| | 2011–2012 | 30 | | | | | Yes | Conn. Gen. Stat. § 12-704e |
| | 2013 | 25 | | | | | Yes | 2017 Conn LS June Sp Sess PA 17–2 (SB 1502) §645 |
| | 2014–2016 | 27.5 | | | | | Yes | |
| | 2017–2020 | 23 | | | | | Yes | |
| DE | 1980–2005 | None | | | | | | |
| | 2006–2020 | 20 | | | | | No | 30 Del. C. § 1117 |
| DC[e] | 1980–1999 | None | | | | | | |
| | 2000 | 10 | | | | | Yes | D.C. Code § 47-1806-04 |
| | 2001–2004 | 25 | | | | | Yes | |
| | 2005–2007 | 35 | | | | | Yes | |
| | 2008–2014 | 40 | | | | | | |
| | 2015–2020 | | 100 | 40 | 40 | 40 | Yes | 2014 DC Laws 20–155 (Act 20–424) Chpt 1C §6(B) |
| FL[b] | 1980–2020 | None | | | | | | |
| GA | 1980–2020 | None | | | | | | |
| HI | 1980–2017 | None | | | | | | |
| | 2018–2020 | 20 | | | | | No | Hawaii Rev Stat. § 235–55.75 |
| ID | 1980–2020 | None | | | | | | |
| IL | 1980–1999 | None | | | | | | |
| | 2000–2002 | 5 | | | | | No | 30 ILCS 5–212 |
| | 2003–2011 | 5 | | | | | Yes | |
| | 2012 | 7.5 | | | | | Yes | |
| | 2013–2016 | 10 | | | | | Yes | |
| | 2017 | 14 | | | | | Yes | 2017 Ill. Legis. Serv. P.A. 100–22 (S.B. 9) |
| | 2018–2020 | 18 | | | | | Yes | 35 ILCS 5–212 |
| IN[f] | 1980–1998 | None | | | | | | |
| | 1999–2002 | | 0 | 3.4 | 3.4 | 3.4 | Yes | Burns Ind. Code Ann. § 6–3.1-21-6 |
| | 2003–2008 | 6 | | | | | Yes | |
| | 2009–2020 | 9 | | | | | Yes | |
| IA[g] | 1980–1989 | None | | | | | | |
| | 1990 | 5 | | | | | No | Iowa Code § 422-12B |
| | 1991–2006 | 6.5 | | | | | No | |
| | 2007–2012 | 7 | | | | | Yes | |
| | 2013 | 14 | | | | | Yes | |
| | 2014–2020 | 15 | | | | | Yes | |

*(Continued)*

**Table 1.** (Continued)

| State | Tax Year | Percent of federal EITC | If different by number of dependents | | | | Refundable | Legal citation |
|---|---|---|---|---|---|---|---|---|
| | | | None[a] | One[a] | Two[a] | Three[a] | | |
| KS | 1980–1997 | None | | | | | | |
| | 1998–2001 | 10 | | | | | Yes | K.S.A. § 79–32,205 |
| | 2002–2006 | 15 | | | | | Yes | |
| | 2007–2009 | 17 | | | | | Yes | |
| | 2010–2012 | 18 | | | | | Yes | |
| | 2013–2020 | 17 | | | | | Yes | |
| KY | 1980–2020 | None | | | | | | |
| LA | 1980–2007 | None | | | | | | |
| | 2008–2018 | 3.5 | | | | | Yes | LA *R.S.* § 47:297.8 |
| | 2019–2020 | 5 | | | | | Yes | Acts 2018, 2nd Ex.Sess., No. 6, § 1 |
| ME | 1980–1999 | None | | | | | | |
| | 2000–2002 | 5 | | | | | No | 36 M.R.S. § 5219-S |
| | 2003–2007 | 4.92 | | | | | No | |
| | 2008 | 5 | | | | | No | |
| | 2009–2010 | 4 | | | | | No | |
| | 2011–2015 | 5 | | | | | No | |
| | 2016–2019 | 5 | | | | | Yes | |
| | 2020 | | 25 | 12 | 12 | 12 | Yes | 2019 Me. Legis. Serv. Ch. 527 (H.P. 1198) (L.D. 1671) |
| MD[g,h] | 1980–1986 | None | | | | | | |
| | 1987–1997 | 50 (NR) | | | | | See Note | Md. TAX-GENERAL Code Ann. § 10–704 |
| | 1998–1999 | 50 (NR) | 0 (R) | 10 (R) | 10 (R) | 10 (R) | See Note | |
| | 2000 | 50 (NR) | 0 (R) | 15 (R) | 15 (R) | 15 (R) | See Note | |
| | 2001–2002 | 50 (NR) | 0 (R) | 16 (R) | 16 (R) | 16 (R) | See Note | |
| | 2003 | 50 (NR) | 0 (R) | 18 (R) | 18 (R) | 18 (R) | See Note | |
| | 2004–2006 | 50 (NR) | 0 (R) | 20 (R) | 20 (R) | 20 (R) | See Note | |
| | 2007–2014 | 50 (NR) 25 (R) | | | | | See Note | |
| | 2015 | 50 (NR) 25.5 (R) | | | | | See Note | |
| | 2016 | 50 (NR) 26 (R) | | | | | See Note | |
| | 2017 | 50(NR) 27(R) | | | | | See Note | |
| | 2018–2020 | 50 (NR) 28(R) | | | | | See Note | 2018 MD Laws Ch. 611 (SB 647) |
| MA | 1980–1996 | None | | | | | | |
| | 1997–2000 | 10 | | | | | Yes | ALM GL ch. 62, § 6 |
| | 2001–2015 | 15 | | | | | Yes | |
| | 2016–2018 | 23 | | | | | Yes | |
| | 2019–2020 | 30 | | | | | Yes | 2018, 154, Sec. 30, 111 |
| MI | 1980–2007 | None | | | | | | |
| | 2008 | 10 | | | | | Yes | MCL § 206.272 |
| | 2009–2011 | 20 | | | | | Yes | |
| | 2012–2020 | 6 | | | | | Yes | |
| MN[g,i] | 1980–1990 | None | | | | | | |
| | 1991–1992 | 10 | | | | | Yes | Minn. Stat. § 290.0671 |
| | 1993–1997 | 15 | | | | | Yes | |
| | 1998 | | 15 | 25 | 30 | 30 | Yes | |
| | 1999 | | 15 | 27 | 32 | 32 | Yes | |
| | 2000–2013 | | 25 | 30 | 35 | 35 | Yes | |

*(Continued)*

**Table 1.** (Continued)

| State | Tax Year | Percent of federal EITC | If different by number of dependents | | | | Refundable | Legal citation |
|---|---|---|---|---|---|---|---|---|
| | | | None[a] | One[a] | Two[a] | Three[a] | | |
| | 2014 | | 26 | 31 | 37 | 37 | Yes | |
| | 2015–2016 | | 26 | 31 | 36 | 36 | Yes | |
| | 2017 | | 25 | 31 | 36 | 36 | Yes | |
| | 2018 | | 25 | 30 | 35 | 35 | Yes | |
| | 2019 | | 53 | 32 | 37 | 38 | Yes | |
| | 2020 | | 52 | 31 | 36 | 38 | Yes | |
| MS | 1980–2020 | None | | | | | | |
| MO | 1980–2020 | None | | | | | | |
| MT | 1980–2018 | None | | | | | | |
| | 2019–2020 | 3 | | | | | Yes | MT ST 15-30-2318; 2017 Montana Laws Ch. 381 (H.B. 391) |
| NE[j] | 1980–2005 | None | | | | | | |
| | 2006 | 8 | | | | | Yes | R.R.S. Neb. § 77-2715-07 |
| | 2007–2020 | 10 | | | | | Yes | |
| NV[b] | 1980–2020 | None | | | | | | |
| NH[b] | 1980–2020 | None | | | | | | |
| NJ | 1980–1999 | None | | | | | | |
| | 2000 | | 0 | 10 | 10 | 10 | Yes | N.J. Stat. § 54A-4-7 |
| | 2001 | | 0 | 15 | 15 | 15 | Yes | |
| | 2002 | | 0 | 17.5 | 17.5 | 17.5 | Yes | |
| | 2003–2006 | | 0 | 20 | 20 | 20 | Yes | |
| | 2007 | 20 | | | | | Yes | |
| | 2008 | 22.5 | | | | | Yes | |
| | 2009 | 25 | | | | | Yes | |
| | 2010–2014 | 20 | | | | | Yes | |
| | 2015 | 30 | | | | | Yes | |
| | 2016–2017 | 35 | | | | | Yes | |
| | 2018 | 37 | | | | | Yes | L.2018, c. 45, § 4 |
| | 2019 | 39 | | | | | Yes | |
| | 2020 | 40 | | | | | Yes | |
| NM | 1980–2006 | None | | | | | | |
| | 2007 | 8 | | | | | Yes | N.M. Stat. Ann. § 7-2-18.15 |
| | 2008–2018 | 10 | | | | | Yes | |
| | 2019–2020 | 17 | | | | | Yes | L. 2019, Ch. 270, §§ 13, 59 |
| NY | 1980–1993 | None | | | | | | |
| | 1994 | 7.5 | | | | | Yes | NY CLS Tax § 606 |
| | 1995 | 10 | | | | | Yes | |
| | 1996–1999 | 20 | | | | | Yes | |
| | 2000 | 22.5 | | | | | Yes | |
| | 2001 | 25 | | | | | Yes | |
| | 2002 | 27.5 | | | | | Yes | |
| | 2003–2020 | 30 | | | | | Yes | |
| NC | 1980–2007 | None | | | | | | |
| | 2008 | 3.5 | | | | | Yes | N.C. Gen. Stat. § 105-151-31 |
| | 2009–2012 | 5 | | | | | Yes | |
| | 2013 | 4.5 | | | | | Yes | |
| | 2014–2020 | None | | | | | | |
| ND | 1980–2020 | None | | | | | | |
| OH | 1980–2012 | None | | | | | | |
| | 2013 | 5 | | | | | No | ORC Ann. § 5747.71 |
| | 2014–2018 | 10 | | | | | No | |
| | 2019–2020 | 30 | | | | | No | 2019 H 62, section 757.100 |

(*Continued*)

**Table 1.** (*Continued*)

| State | Tax Year | Percent of federal EITC | If different by number of dependents | | | | Refundable | Legal citation |
|---|---|---|---|---|---|---|---|---|
| | | | None[a] | One[a] | Two[a] | Three[a] | | |
| OK | 1980–2001 | None | | | | | | |
| | 2002–2015 | 5 | | | | | Yes | 68 Okl. St. § 2357–43 |
| | 2016–2020 | 5 | | | | | No | Laws 2016, c. 341, § 1; |
| | | | | | | | | OK ADC 710:50-15-90 |
| OR[k] | 1980–1996 | None | | | | | | |
| | 1997–2005 | 5 | | | | | No | ORS § 315.266 |
| | 2006–2007 | 5 | | | | | Yes | |
| | 2008–2013 | 6 | | | | | Yes | |
| | 2014–2019 | 8 | | | | | Yes | ORS §§ 315.266(a), (b); |
| | | | | | | | | Laws 2016, c. 98, § 1 |
| | 2020 | 9 | | | | | Yes | Laws 2019, c. 579, §§ 31, 32a |
| PA | 1980–2020 | None | | | | | | |
| RI[l] | 1980–2000 | None | | | | | | |
| | 2001 | 25.5 | | | | | No | R.I. Gen. Laws § 44-30-2.6 |
| | 2002 | 25 | | | | | No | |
| | 2003–2014 | 25 | | | | | Partial Refund | |
| | 2015 | 10 | | | | | Yes | P.L. 2014, ch. 145, art. 12, §§ 7, 22 |
| | 2016 | 12.5 | | | | | Yes | |
| | 2017–2020 | 15 | | | | | Yes | P.L. 2016, ch. 142, art. 13, § 15, 20 |
| SC | 1980–2017 | None | | | | | | |
| | 2018 | 20.83 | | | | | No | SC ST § 12-6-3632; 2017 Act No. 40 (H.3516), § 16.A. |
| | 2019 | 41.66 | | | | | No | |
| | 2020 | 62.49 | | | | | No | |
| SD[b] | 1980–2020 | None | | | | | | |
| TN[b] | 1980–2020 | None | | | | | | |
| TX[b] | 1980–2020 | None | | | | | | |
| UT | 1980–2020 | None | | | | | | |
| VT | 1980–1987 | None | | | | | | |
| | 1988 | 23 | | | | | Yes | 32 V.S.A. § 5828b |
| | 1989–1990 | 25 | | | | | Yes | |
| | 1991–1993 | 28 | | | | | Yes | |
| | 1994–1999 | 25 | | | | | Yes | |
| | 2000–2017 | 32 | | | | | Yes | |
| | 2018–2020 | 36 | | | | | Yes | 2017, Adj. Sess., Sp. Sess., No. 11, §§ H.4, H.31 |
| VA[m] | 1980–1999 | None | | | | | | |
| | 2000–2005 | Other | | | | | | |
| | | See Notes | | | | | | |
| | 2006–2020 | 20 | | | | | No | Va. Code Ann. § 58.1–339.8 |
| WA[b,n] | 1980–2007 | None | | | | | | |
| | 2008–2009 | 5 | | | | | Yes | Rev. Code Wash. (ARCW) § 82.08.0206 |
| | 2010–2020 | 10 | | | | | Yes | |
| WV | 1980–2020 | None | | | | | | |
| WI[o] | 1980–1983 | None | | | | | | |
| | 1984–1985 | | 0 | 30 | 30 | 30 | No | Wis. Stat. § 71.09 |
| | 1986–1988 | None | | | | | | |
| | 1989–1993 | | 0 | 5 | 25 | 75 | Yes | Wis. Stat. § 71.07 |
| | 1994 | | 0 | 1.15 | 6.25 | 18.75 | Yes | |
| | 1995 | | 0 | 4 | 16 | 50 | Yes | |
| | 1996–2010 | | 0 | 4 | 14 | 43 | Yes | |
| | 2011–2020 | | 0 | 4 | 11 | 34 | Yes | |

(*Continued*)

**Table 1.** (Continued)

| State | Tax Year | Percent of federal EITC | If different by number of dependents | | | | Refundable | Legal citation |
|---|---|---|---|---|---|---|---|---|
| | | | None[(a)] | One[(a)] | Two[(a)] | Three[(a)] | | |
| WY[(b)] | 1980–2020 | None | | | | | | |

Notes:

NR—Non-Refundable.

R—Refundable.

(a) Variation from federal EITC percentage if different by number of dependents.

(b) No state earned income tax.

(c) California uses a different income eligibility requirement than the federal EITC, imposing a lower maximum income threshold to qualify for the credit; the rates listed in Table 1 are effective percentages of the federal EITC, calculated to account for the differences between the California and federal EITC. The California EITC had a minimum age requirement of 25 years old for taxpayers without dependents from 2015–2017; the minimum age was lowered to 18 years old effective in tax year 2018. Beginning in 2019, taxpayers eligible for EITC under Section 17052 who had one or more children under the age of 6 were also allowed a young child tax credit with a maximum benefit of $1,000 and a phase-out threshold of $25,000. Cal Rev & Tax Code § 17052.1; added by CA Stats. 2019, c. 39 (AB 91).

(d) The Colorado EITC was established as a refund mechanism under CO's Taxpayer Bill of Rights (TABOR) and, prior to 2016, required a budget surplus for the EITC credit to be financed. Thus, in some years tax payers did not receive the credit (*). From 2005 to 2010, TABOR was temporarily suspended by referendum, effectively suspending the EITC. In 2013, a law was passed establishing a separate permanent EITC law that would be available the year after the next year in which an EITC refund would be triggered under TABOR. This occurred in 2015 with the permanent EITC becoming available starting in 2016. *No surplus was available from 2002 to 2004 or from 2011 to 2014, and the EITC was unfunded during these periods. As a TABOR refund mechanism, the EITC was funded only from 1999–2001.

(e) DC taxpayers 25 years and older with no dependents were allowed a credit that is 100% of the federal EITC beginning in 2015. The DC phase-out threshold and completed phase-out amounts are greater than the respective federal amounts, which means that the effective percentage for these taxpayers can exceed 100% of the federal.

(f) Indiana specifies a fixed dollar amount with a maximum credit of $408 from 1999 to 2002. The percentage shown is calculated based on the federal amount for those years.

(g) The majority of states use the federal EITC eligibility requirements to determine whether an individual is eligible for state EITC credits. As such, it is important to note that childless adults were not eligible for the federal EITC prior to the passage of the Omnibus Budget Reconciliation Act of 1993 (OBRA93). Thus, while IA, MD, MN, and VT did not differentiate the state credit percentage based on the number of dependents, it would not be applicable to childless adults. Once OBRA93 became fully effective in 1994, childless adults between the ages of 25 and 65 years old are eligible for state credits unless they have state-specific eligibility requirements.

(h) Maryland residents calculate a nonrefundable credit, which is equal to the lesser of 50% of the federal credit or the state income tax liability in the taxable year. In 1998, a refundable tax credit was also implemented. Individuals can select either the refundable or the non-refundable credit, but not both. However, from 1998 to 2006, an individual had to have one or more qualifying dependents to have the refundable credit as an option. In 2018, qualified adult taxpayers under the age of 25 without dependents became eligible for the Maryland EITC.

(i) Minnesota's Working Family Credit was specified as a percentage of the federal EITC from 1991–1997. Starting in 1998, the credit was restructured as a percentage of earnings, with maximum credits and income eligibility requirements specified. We use these maximum credits and the respective income eligibility requirements to calculate the effective maximum percent of federal EITC. Values in table reflect this maximum. Implemented in 2019, Minnesota created a separate category for taxpayers with three or more children and lower the age of eligibility for taxpayers with no qualifying children to 21 years of age.

(j) Effective in 2015, Nebraska modified its EITC so that net operating loss carryforward must be added back in as income when qualifying for the EITC.

(k) Beginning in 2017, Oregon implemented an additional child tax credit worth 3% of the federal EITC. To qualify for this credit, you must qualify for the federal EITC as well as have a qualifying dependent under 3 years of age.

(l) Rhode Island had a partially refundable EITC from 2003 to 2014. The policy allows a percentage of the amount exceeding the tax liability to be refundable. For years 2003 and 2004, the percentage is 5. For 2005, the percentage was increased to 10. Then from 2006 to 2014, the percentage was 15.

(m) Virginia also has a direct amount available that begin on January 1, 2000. The eligibility for the direct amount differs from the federal EITC eligibility by imposing a lower maximum income threshold. Additionally, the number of direct amount credits that can be claimed is based on the number of qualified dependents and whether a spouse is on the tax return. The credit based on the federal amount and eligibility requirements began on January 1, 2006. When both of these options are in effect (January 1, 2006 and onward), an individual decides which of the amounts to utilize.

(n) Washington passed legislation to create a state-based EITC in 2009 with an automatic rate increase in 2010. However, this policy was never implemented because lawmakers have not financed the credit. Additionally, Washington has a direct amount of $25 from 2008 to 2010 and $50 from 2011 to 2016, with the individual receiving the greater of the direct amount or refund percentage.

(o) Wisconsin's credit in 1994 is not based on the federal credit. Maximum credits are specified. We use these maximum credits to calculate the maximum percent of federal EITC. Values in table reflect this effective maximum for 1994 only. All other years are percent of federal EITC.

percent of federal EITC), variation by number of children in the household, refundability, and, finally, specific legal citations (see S3 and S4 Appendices for data codebook and cvs data file, respectively). As of 2020, 29 states—as well as Washington D.C.—had a state EITC. Nine states do not have an earned income tax (Alaska, Florida, Nevada, New Hampshire, South Dakota, Tennessee, Texas, Washington and Wyoming). In 2020, the majority of states with an EITC calculate the state credit as a percent of the federal EITC (27 of 29 states) and offer the state EITC as a refundable tax credit (23 of 29 states). Being refundable means that if the EITC reduces the taxpayer's tax liability to zero, any remaining credit amount will be refunded to the taxpayer, usually in the form of a refund check sent from the state to the taxpayer [48]. Six state EITCs are nonrefundable credits, which provide no further income benefit beyond a zero tax liability [48]. Although the state of Washington does not have an income tax, it does have an EITC law. Enacted in 2008 as a refundable credit, Washington state's EITC has not yet been implemented as policymakers have not financed the credit.

## Trends over time

Fig 1 shows cumulative EITC policy adoption over time. The pattern roughly follows the well-known S-curve of diffusion [19, 49] -a slow start with 5 states adopting from the mid-1980s to mid-1990s, followed by a more rapid increase in state adoptions from the mid-1990s to 2010 (19 additional jurisdictions), followed by a decrease in rate of additional adoptions over the 2010–2020 decade (ending with 30 jurisdictions having an EITC). Given that remaining states continue to introduce EITC legislation, it appears that the curve is not yet complete, and it is not yet known the degree to which the next decade will follow the traditional S-curve expectation of further slowing of the adoption rate (i.e., among "laggard" states). Geographic diffusion of state EITC legislation from 1993 (following the major expansion of the federal EITC in the Omnibus Budget Reconciliation Act of 1993) through 2020 is illustrated in Fig 2, with a series of maps of state EITC features in 1993, 2000, 2010, and 2020. Only five states had an EITC prior to 1993: IA, MD, MN, VT, and WI (see Table 1). In 1993, three had a refundable EITC. Iowa and Maryland had a nonrefundable EITC. By 2000, eleven additional states introduced an EITC, seven of which were refundable. These state credits ranged widely in magnitude, from 3.4% (IN for one or more dependents) to 43% (WI for three dependents) of the federal amount for refundable credits, and 5% (IL, ME, OR) to 50% (MD nonrefundable option) for nonrefundable credits. By 2010, two additional states introduced nonrefundable credits (DE, RI) (At that time, Rhode Island's EITC was partially refundable. See Table 1, note l for discussion.) and seven introduced refundable credits. In addition, the three states that had had nonrefundable tax credits changed them to refundable (IA, IL, OR). By 2020, two (ME, RI) of the four nonrefundable state EITCs became refundable and two additional states (OH, HI) introduced a nonrefundable EITC. An additional three states introduced a refundable credit (CA, CT, MT), Oklahoma changed its refundable EITC to a nonrefundable credit, and North Carolina eliminated its EITC. In 2020, state credits ranged in size from 3% (MT) to 45% (CA) and 52% (MN with no dependents) of the federal EITC for refundable credits, and 20% (HI) to 62% (SC) for nonrefundable credits.

Fig 3 presents EITC levels, refundability, and legislative changes from 1993 through 2020 by state and year. From 1993 through 2020, 114 state EITC legislative changes occurred. Notably active years were in 2000 (n = 9 changes), 2008 (n = 8 changes), and 2019 (n = 8 changes).

In 1993 five states had an EITC (IA, MN, WI, VT, MD). Between 1993 and 2020, an additional 24 states and the District of Columbia introduced an EITC and 28 states amended EITC legislation. Most amendments increased the credit (i.e., increased percent of the federal EITC) (n = 18: DC, CA, CO, IA, IL, IN, KS, MA, MN, NE, NJ, NM, NY, OH, OR, SC, VT, WA) and/ or changed from a nonrefundable to a refundable credit (n = 6: IA, IL, ME, OR, RI, WI).

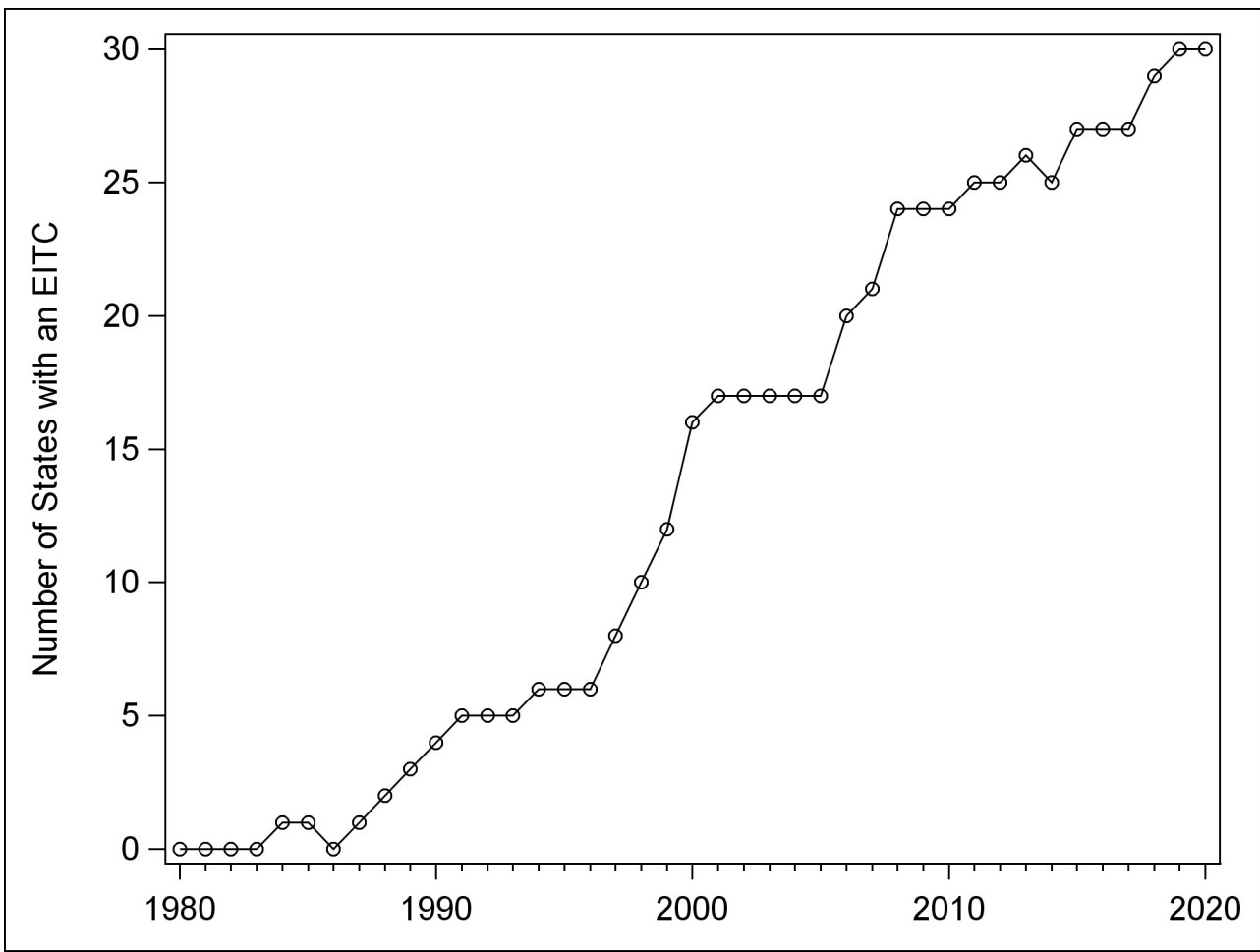

**Fig 1. U.S. states adoption of EITC by year.**

There are also a handful of states that have reduced their EITC credit, removed refundability, or eliminated their EITC altogether. For example, New Jersey incrementally increased their credit to 25% in 2009, then decreased it to 20% from 2010–2014, and then incrementally increased it to 40% in 2020. Connecticut's credit was introduced in 2011 at 30%, then was reduced to 25% in 2013, followed by an increase to 27.5% for 2014 to 2016, followed by a reduction to 23% from 2017–2020. Rhode Island reduced the credit from 25.5% in 2001 to 15% in 2020 along with a transition from nonrefundable to refundable credit during the period. Michigan and Wisconsin decreased the amount of their refundable credit.

## Regional variations

Three of the five early adopter states were in the upper Midwest. States in the upper Midwest and Northeast have implemented the most legislative changes, and have had the highest credit amounts, especially the states in the Northeast. Over 75% of states in the Midwest and Northeast have state EITCs, compared with around 40% of states in the West and South. In 2020, the highest state EITCs, as a percent of the federal EITC amount, were in Washington, D.C. (40%), California (43%), and in Minnesota for those with no dependents (52%).

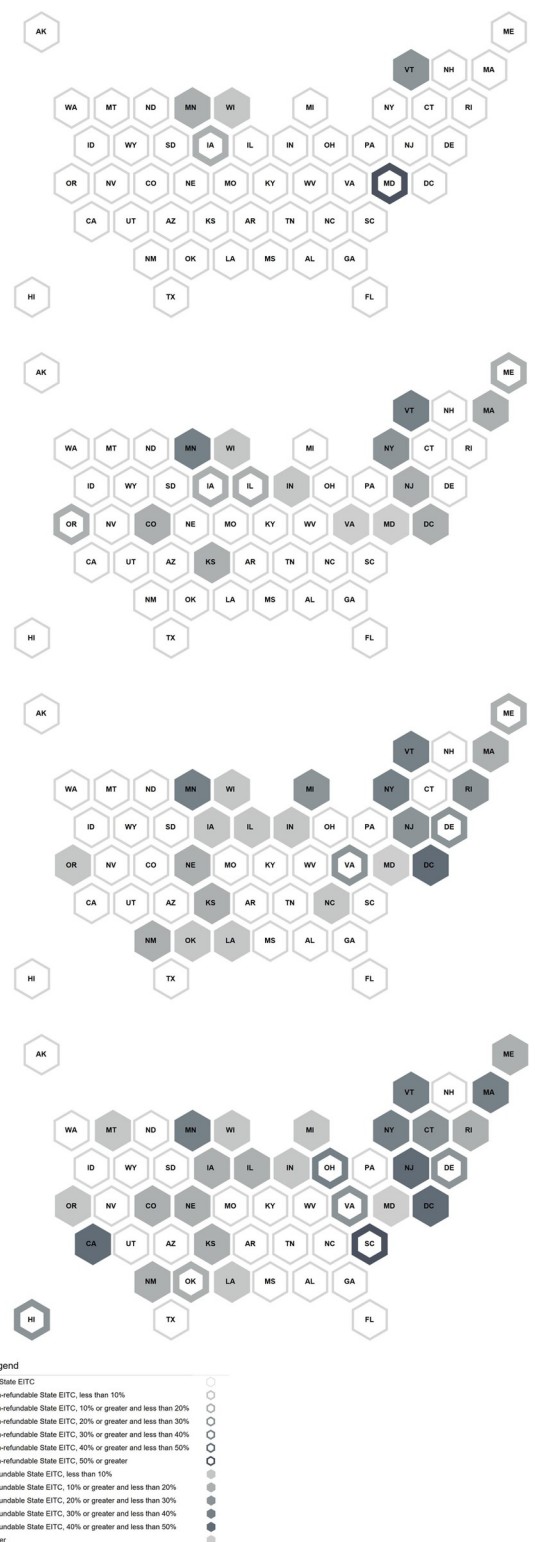

**Fig 2. Maps of state EITC credits and refundability based on having one dependent: 1993, 2000, 2010, 2020.**

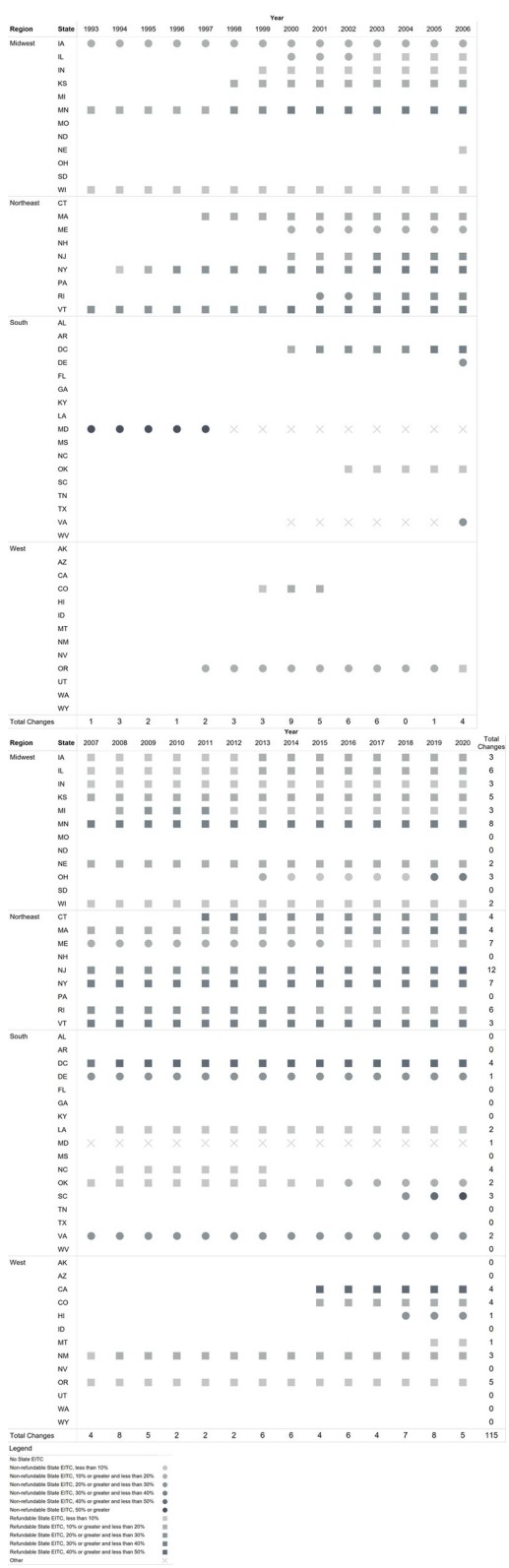

**Fig 3. Trend of state EITC credits and refundability based on one dependent: 1993–2020.**

## Discussion

Our legal research documents the diffusion of EITC legislation across states and time from 1980 to 2020. Since the late 1980s, there has been a gradual increase in policy adoption with 29 of 50 states and Washington D.C. having adopted a state EITC by 2020. Three key findings inform future research and policy development in this area. First, the pattern of diffusion across states and time shows initial introductions during the 1990s in the Midwest, then spreading to the Northeast, with more recent expansions in the West and South. A few states initially began to pass EITC legislation during the 1980s. By 1993, when the major expansion of the federal EITC occurred as part of welfare reform, only five states had an EITC, three of which were in the upper Midwest. After the relatively slow start, the late 1990s through 2008 was a particularly active period for adoption of a state-level EITC.

By 2000 an additional eleven states introduced an EITC bringing the total to 16 states. Most of this wave of state EITC implementation occurred in the Northeast and Midwest. From 2001 to 2010 another eight states introduced an EITC, bringing the total to 24. Again, uptake occurred largely in the Northeast and Midwest, with credit enactment also beginning to spread to the West and South. Following the major waves of expansion prior to 2010, the pace of state EITC introductions slowed.

From 2010 to 2020, seven states adopted an EITC, and one state eliminated its EITC, bringing the total, as of 2020, to 29 states and D.C., including Washington state, which has not yet implemented its EITC. The majority of states in the Midwest and Northeast and over a third of states in the West and South now have an EITC. As noted, the overall diffusion pattern of this public policy roughly approximates the conventional S-curve of cumulative adoptions over time—a few innovator states slowly start the process, followed by a period of more rapid spread, ending with a gradually slowing spread to the remaining laggard states. Given that a number of states have enacted EITC laws in the past few years (Table 1), combined with the fact that 40% of states do not yet have a state-level EITC, suggests that we likely have not yet reached the top of the S-curve—that is, a complete leveling off of the adoption rate [49, 50].

Differences across state and time of fundamental policy dimensions are evident, including size of credit and refundability. As of 2020, the 29 states and D.C. with an EITC varied widely in credit amount (5% to 52% of the federal EITC) and varied slightly in refundability (23 of 29 were refundable). Following the introduction of an EITC, about half of the states continued to increase the credit amount and/or convert a nonrefundable credit to a refundable credit. About a third of the states have made adjustments to increase or decrease credits throughout the years. One state recently eliminated its EITC.

State EITC benefits vary considerably by household structure. Most state EITCs are based on the federal EITC. The federal EITC was designed to incentivize work, especially for households with children [20]. EITCs for individuals or couples with no children are limited. For example, based on 2019 U.S. federal poverty thresholds, the 2019 federal EITC credit is nearly phased out for a household with no children at the poverty threshold (e.g., household with one adult with an income of $12,490 would receive a $237 refund; household with two adults and no dependents with an income of $16,910 would receive a $340 refund). In comparison, households with children with income at the poverty level would receive near the maximum EITC (e.g., household with one adult and two dependents with an income of $21,330 would receive a $5,345 credit; household with two adults and two dependents with an income of $25,750 would receive a $5,627 refund) [51]. These numbers translate to a 2% boost in income among households with no children compared with a 22% to 25% boost for households with two children. Nearly all state EITCs are based on the federal eligibility and income requirements; Minnesota and Wisconsin, however, use different income thresholds, which results in

higher state credits to households with children. Consistent with how EITC benefits are structured, population segments that receive larger credits are those found to benefit the most in regard to health and well-being, especially among infants and children [23, 34, 37–43].

A broader question that arises whenever observing the spread of a particular state policy is: What drives the adoption of this distinct policy in a given state at a given time, and what drives the pattern of diffusion to other states? Understanding the complex mix of causal factors affecting state-level policy adoption across time is of theoretical and practical importance—and is an important focus of current and future research in political science [50]. There is a wide range of conceptually distinct plausible drivers of policy adoption [50]. In broad categories these drivers include, at a minimum, geospatial effects, role of multiple dimensions of political and social ideologies, effects of coercion and incentives (from higher level federal government, as one key example), and resource availability and distribution [50]. Moreover, recent research suggests drivers of policy adoption likely vary in their relevance across the adoption curve (i.e., across time within each adoption curve) [50]. There remains considerable poorly understood heterogeneity in policy adoption speed, and continuing research on causes of adoption patterns, both for EITC and more generally for all state-level policies, is warranted.

Separate from such research on policy adoption processes, research continues to accumulate regarding the societal benefits of EITCs, including reductions in family poverty and improved health outcomes [23, 33–41]. The analysis of policy trends reported here extends to 2020. Our research highlights that legislative debate and action at the state level has recently been and continues to be active, underscoring the utility of continued policy surveillance and public health law research evaluating health and well-being effects of new credits and other legislative changes to tax credit law. Additionally, cost-benefit analysis of state EITCs would also be helpful, given consistent findings of beneficial effects on birth weight and gestation weeks, and the annual social and health costs of preterm or low-birth-weight births in the U.S [52].

## Conclusion

State-level EITC legislation continues to diffuse across states, with the majority of states in the Midwest and Northeast and over one-third of states in the West and South having adopted state EITCs as of 2020. States continue to modify their laws, with most increasing the EITC benefits over time. Appendices to this paper provide the policy surveillance codebook, the coding protocol, the data codebook, and a cvs data file ready for further analyses. We hope these resources support future researchers conducting multistate and multi-policy studies of predictors of policy adoption and evaluations of the effects of tax credit laws on the wide range of potential health and education outcomes. Finally, we hope this collaboration—between lawyers, policy analysts, epidemiologists, and economists—serves as an example for high-quality empirical studies of other dimensions of policy and law with the potential to affect the social determinants of health.

## Supporting information

**S1 Appendix. LawAtlas EITC codebook.**
(PDF)

**S2 Appendix. LawAtlas EITC research protocol.**
(DOCX)

**S3 Appendix. EITC data codebook 2020.**
(DOCX)

**S4 Appendix. CSV data file.**
(CSV)

**S5 Appendix. CDC EITC research protocol.**
(DOCX)

## Acknowledgments

The authors thank Scott Burris, JD, and Lindsay K. Cloud, JD, with Temple University Beasley School of Law Center for Public Health Law Research and the LawAtlas team for legal scholarship and coding through 2016.

## Author Contributions

**Conceptualization:** Kelli A. Komro, Phenesse Dunlap.

**Data curation:** Phenesse Dunlap, Nolan Sroczynski, Melvin D. Livingston, Megan A. Kelly, Dawn Pepin.

**Formal analysis:** Nolan Sroczynski, Melvin D. Livingston, Sara Markowitz.

**Funding acquisition:** Kelli A. Komro, Alexander C. Wagenaar.

**Investigation:** Kelli A. Komro.

**Methodology:** Phenesse Dunlap.

**Project administration:** Shelby Rentmeester.

**Validation:** Sara Markowitz.

**Visualization:** Nolan Sroczynski.

**Writing – original draft:** Kelli A. Komro, Phenesse Dunlap, Nolan Sroczynski, Melvin D. Livingston, Sara Markowitz, Shelby Rentmeester.

**Writing – review & editing:** Kelli A. Komro, Phenesse Dunlap, Nolan Sroczynski, Melvin D. Livingston, Megan A. Kelly, Dawn Pepin, Sara Markowitz, Shelby Rentmeester, Alexander C. Wagenaar.

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
