## [Decision Letter · Decision Letter 0]

25 Sep 2020

PONE-D-20-25783

Anti-poverty policy and health: Policy diffusion of state Earned Income Tax Credits across the U.S. states from 1980 to 2020

PLOS ONE

Dear Dr. Komro,

Thank you for submitting your manuscript to PLOS ONE. After careful consideration, we feel that it has merit but does not fully meet PLOS ONE’s publication criteria as it currently stands. Please especially note reviewer two's comments.  Therefore, we invite you to submit a revised version of the manuscript that addresses the points raised during the review process.

We look forward to receiving your revised manuscript.

Kind regards,

Sze Yan Liu, PhD

Academic Editor

PLOS ONE

Journal Requirements:

'The National Institute on Minority Health and Health Disparities, National Institutes of Health (https://www.nimhd.nih.gov) through award R01MD010241 to KAK and ACW and the Policy Research, Analysis, and Development Office; Office of the Associate Director for Policy and Strategy; Centers for Disease Control and Prevention (https://www.cdc.gov/policy/about/index.html) supported this research. The content is solely the responsibility of the authors and does not necessarily represent the views of the National Institutes of Health or the Centers for Disease Control and Prevention. The funders had no role in study design, data collection and analysis, decision to publish, or preparation of the manuscript. '

We note that one or more of the authors are employed by a commercial company: RTI International

Reviewers' comments:

Reviewer's Responses to Questions

**Comments to the Author**

1. Is the manuscript technically sound, and do the data support the conclusions?

Reviewer #1: Yes

Reviewer #2: Partly

2. Has the statistical analysis been performed appropriately and rigorously? 

Reviewer #1: Yes

Reviewer #2: Yes

3. Have the authors made all data underlying the findings in their manuscript fully available?

Reviewer #1: Yes

Reviewer #2: Yes

4. Is the manuscript presented in an intelligible fashion and written in standard English?

Reviewer #1: Yes

Reviewer #2: Yes

5. Review Comments to the Author

Reviewer #1: Congratulations to the authors for the great work! It is very important to study policy diffusion while such study is limited, especially in public health. I found that this initial study on Policy diffusion of state Earned Income Tax Credits (EITC) will be beneficial for other works to evaluate and assess the impacts in population.. Authors has also created a clear structure, so the manuscript is well-written and precise.

I do not have major comments for this manuscript, but there are few comments for the authors.

1. The authors use maps to describe the diffusion. When authors describe how the EITC policies diffused over 40 years and explain the diffusion within the cardinal direction, I wonder whether it diffused through adjacent states or not. I need to ask this as studying the policy diffusion is very important to incorporate how adjacent area stimulates the policy diffusion while the authors do not consider the vertical diffusion pattern in the study. For me, this is very important for international readers and experts to understand the geographical factors in the EITC policy diffusion aside from the cardinal direction. Perhaps the authors may consider to add one or two sentences explaining the adjacent states’ influence in the diffusion.

2. “The overall diffusion pattern of this public policy roughly approximates the conventional S-curve of diffusion—a few innovator states slowly start the diffusion process, followed by a period of rapid spread, ending with a gradually slowing spread to the remaining laggard states.” Just to make this sentence stronger, perhaps the authors can explain more about this in the results and add the period/year for each phase to explain when the innovators, early adopters, etc began. This claim seems very weak without a support explaining each phase.

Reviewer #2: This article looks at the diffusion of the earned income tax credit from 1980-2020, with an eye toward out the geographic locations of the diffusion. While this article is fundamentally sound in its methodological approach, I see no theory (or even postulation) about why these policies are diffusing from one point to the next. Are states learning from one another, emulating other’s policies? Is this competition? A statement needs to be made regarding why states are adopting this specific policy, and how this diffusion (the spreading of the policy from one jurisdiction to the next) is occurring across states. Since you are looking at just adoption years with no modeling to connect sates to one another (many studies rely on survival analyses), I would at least postulate a driving mechanism for states adopting this policy. I would also consider doing a cumulative adoption graph and looking at how many adoptions occurred per year. Also, I would look at the newer literature relevant to diffusion research.

6. PLOS authors have the option to publish the peer review history of their article (what does this mean?). If published, this will include your full peer review and any attached files.

Reviewer #1: No

Reviewer #2: **Yes: **Joshua L Mitchell

---

## [Author Response · Author response to Decision Letter 0]

16 Oct 2020

PONE-D-20-25783

Anti-poverty policy and health: Policy diffusion of state Earned Income Tax Credits across the U.S. states from 1980 to 2020

PLOS ONE

Sze Yan Liu, PhD

Academic Editor

PLOS ONE

Dear Dr. Sze Yan Liu:

Thank you for the opportunity to revise and resubmit our manuscript for consideration of publication in PLOS ONE. We have carefully reviewed the comments and revised the paper accordingly. Below we have copied each point raised by the editor and reviewers and summarize our response with italicized text following each point. We have also included a marked-up and unmarked versions of the manuscript.

Journal Requirements:

1. We have followed the PLOS ONE’s style requirements.

2. We have revised our Financial Disclosure statement. 

We originally listed RTI International (https://www.rti.org), a nonprofit research institute, as an affiliation for Nolan Sroczynski. However, this research was conducted while Mr. Sroczynski was a student at Emory University Rollins School of Public Health. While Mr. Sroczynski is currently employed at RTI International, his involvement in this research occurred while he was a student at Emory and recently during his own time. RTI International did not provide any support for this research, even in the way of overhead salary. Therefore, we feel it is more appropriate to remove the RTI affiliation for him.

Phenesse Dunlap receives funding from a training grant and we would like to include this source of funding in our financial disclosure section. 

The revised Financial Disclosure section:

'The National Institute on Minority Health and Health Disparities, National Institutes of Health (https://www.nimhd.nih.gov) through award R01MD010241 to KAK and ACW, The National Heart, Lung, and Blood Institute, National Institutes of Health (https://www.nhlbi.nih.gov) through training award T32 HL130025 to PD, and the Policy Research, Analysis, and Development Office; Office of the Associate Director for Policy and Strategy; Centers for Disease Control and Prevention (https://www.cdc.gov/policy/about/index.html) supported this research. The findings and conclusions of this paper are solely the responsibility of the authors and do not necessarily represent the official position of the National Institutes of Health or the Centers for Disease Control and Prevention. The funders had no role in study design, data collection and analysis, decision to publish, or preparation of the manuscript. '

3. We have included captions for your Supporting Information files at the end of our manuscript, and update any in-text citations to match accordingly.

4. We inserted our ethics statement in the Methods section of our manuscript. 

Reviewers' comments:

Reviewer #1: Congratulations to the authors for the great work! It is very important to study policy diffusion while such study is limited, especially in public health. I found that this initial study on Policy diffusion of state Earned Income Tax Credits (EITC) will be beneficial for other works to evaluate and assess the impacts in population. Authors has also created a clear structure, so the manuscript is well-written and precise.

Thank you for your positive feedback.

I do not have major comments for this manuscript, but there are few comments for the authors.

1. The authors use maps to describe the diffusion. When authors describe how the EITC policies diffused over 40 years and explain the diffusion within the cardinal direction, I wonder whether it diffused through adjacent states or not. I need to ask this as studying the policy diffusion is very important to incorporate how adjacent area stimulates the policy diffusion while the authors do not consider the vertical diffusion pattern in the study. For me, this is very important for international readers and experts to understand the geographical factors in the EITC policy diffusion aside from the cardinal direction. Perhaps the authors may consider to add one or two sentences explaining the adjacent states’ influence in the diffusion.

Thank you for your thoughtful comment. Understanding the complex mix of causal factors driving state-level policy adoption across time is of theoretical and practical importance—and is an important focus of current and future research in political science. There is a wide range of conceptually distinct plausible drivers of policy adoption. In broad categories they include, at a minimum, geospatial effects, role of multiple dimensions of political and social ideology, effects of coercion and incentives (from higher level federal government, as one key example), and resource availability and distribution issues. Moreover, recent research suggests drivers of policy adoption likely vary in their importance across the adoption curve (i.e., across time within each adoption curve). Additionally, there remains considerable poorly understood heterogeneity in adoption speed. A study with the goal of advancing understanding of the relative role of factors affecting policy adoption across states would require study of dozens or hundreds of policy adoptions; a study of one policy is essentially an n=1 study in such a context. Our study is a descriptive study of EITC adoption and EITC policy content across states; the objective is describing the diffusion of this one policy and, most importantly, providing the requisite datasets for further empirical evaluations of the health effects of this particular economic security policy. Table 1 provides the complete clean policy dataset, and Figure 1 provides maps illustrating the spatial spread of adoption over time. To better address these broader conceptual issues raised by reviewer 1 we have: (1) added to the Discussion section text on the causes of policy diffusion and how that is one direction for future research, and (2) added a new Figure 1 showing the cumulative policy diffusion curve for this policy, and added description of that curve in more detail. The figure provides data from this single additional policy case of interest to the general political science literature on factors affecting diffusion, while also showing the data behind our statement that EITC policy diffusion roughly appears to be following the standard S-curve of innovation diffusion. 

2. “The overall diffusion pattern of this public policy roughly approximates the conventional S-curve of diffusion—a few innovator states slowly start the diffusion process, followed by a period of rapid spread, ending with a gradually slowing spread to the remaining laggard states.” Just to make this sentence stronger, perhaps the authors can explain more about this in the results and add the period/year for each phase to explain when the innovators, early adopters, etc began. This claim seems very weak without a support explaining each phase.

This is a good idea. We have added a new Figure 1, illustrating cumulative adoptions over time, and describe period/years for each phase in the text. We introduce this figure in the beginning of the Time Trends in the Results section. Then Figure 2 and 3 follow, providing additional details of adoptions over time, with the details summarized in the Results and Discussion section—these are helpful to forward the primary objective of the paper—fostering additional studies of the health effects of EITC and similar policies

Reviewer #2: This article looks at the diffusion of the earned income tax credit from 1980-2020, with an eye toward out the geographic locations of the diffusion. While this article is fundamentally sound in its methodological approach, I see no theory (or even postulation) about why these policies are diffusing from one point to the next. Are states learning from one another, emulating other’s policies? Is this competition? A statement needs to be made regarding why states are adopting this specific policy, and how this diffusion (the spreading of the policy from one jurisdiction to the next) is occurring across states. Since you are looking at just adoption years with no modeling to connect sates to one another (many studies rely on survival analyses), I would at least postulate a driving mechanism for states adopting this policy. I would also consider doing a cumulative adoption graph and looking at how many adoptions occurred per year. Also, I would look at the newer literature relevant to diffusion research.

 Thank you for your thoughtful suggestions.

1. As noted in response to reviewer 1 above, we have added a new Figure 1 presenting cumulative adoptions over time, and describe the pattern in the text. We present this figure in the beginning of the Time Trends in the Results section on page 7. 

2. We added a citation to a great 2020 publication specifically focused on this issue of the causal drivers of state-level policy adoption (Mallinson, D.J. (2020). Policy innovation and adoption across the diffusion like course. Policy Studies Journal, 1-24. doi: 10.1111/psj.12406). We added text to the Discussion section on the many possible factors influencing policy adoption. 

3. We also suggest future research to study predictors of policy adoption. In response to reviewer 1 we point out that our study was not designed to answer that question, and a fundamentally different study design would be required to substantively contribute to the political science literature on drivers of state policy diffusion. Our goal was to describe state-level EITC policy spread and provide a complete policy dataset, for the purpose of facilitating additional empirical research evaluating the health effects of EITC and similar policies that directly address family economic security.

4. We changed the title to better encompass the study purpose and lesson the sole focus on the term “diffusion.”

---

## [Decision Letter · Decision Letter 1]

4 Nov 2020

Anti-poverty policy and health: Attributes and diffusion of state Earned Income Tax Credits across U.S. states from 1980 to 2020

PONE-D-20-25783R1

Dear Dr. Komro,

We’re pleased to inform you that your manuscript has been judged scientifically suitable for publication and will be formally accepted for publication once it meets all outstanding technical requirements.

Kind regards,

Sze Yan Liu, PhD

Academic Editor

PLOS ONE

Additional Editor Comments (optional):

The additional revisions and clarification greatly strengthens your paper.

Reviewers' comments:

Reviewer's Responses to Questions

**Comments to the Author**

1. If the authors have adequately addressed your comments raised in a previous round of review and you feel that this manuscript is now acceptable for publication, you may indicate that here to bypass the “Comments to the Author” section, enter your conflict of interest statement in the “Confidential to Editor” section, and submit your "Accept" recommendation.

Reviewer #2: All comments have been addressed

2. Is the manuscript technically sound, and do the data support the conclusions?

Reviewer #2: Yes

3. Has the statistical analysis been performed appropriately and rigorously? 

Reviewer #2: Yes

4. Have the authors made all data underlying the findings in their manuscript fully available?

Reviewer #2: Yes

5. Is the manuscript presented in an intelligible fashion and written in standard English?

Reviewer #2: Yes

6. Review Comments to the Author

Reviewer #2: The author/s addressed all of my comments pretty well. I look forward to seeing this article in print. I would recommends the author/s give the paper a final edit before submission.

7. PLOS authors have the option to publish the peer review history of their article (what does this mean?). If published, this will include your full peer review and any attached files.

Reviewer #2: **Yes: **Joshua L Mitchell

---

## [Editor Report · Acceptance letter]

9 Nov 2020

PONE-D-20-25783R1 

Anti-poverty policy and health: Attributes and diffusion of state Earned Income Tax Credits across U.S. states from 1980 to 2020 

Dear Dr. Komro:

I'm pleased to inform you that your manuscript has been deemed suitable for publication in PLOS ONE. Congratulations! Your manuscript is now with our production department. 

Kind regards, 

on behalf of

Dr. Sze Yan Liu 

Academic Editor

PLOS ONE